# Botulinum Toxin Type A to Improve Facial Symmetry in Facial Palsy: A Practical Guideline and Clinical Experience

**DOI:** 10.3390/toxins13020159

**Published:** 2021-02-18

**Authors:** Carla de Sanctis Pecora, Danielle Shitara

**Affiliations:** Independent Researcher, São Paulo 04029-200, Brazil; danielleshitara@yahoo.com.br

**Keywords:** facial palsy, facial paralysis, neurotoxin, botulinum toxin A, incobotulinum-toxin A, facial asymmetry

## Abstract

Unilateral peripheral facial nerve palsy jeopardizes quality of life, rendering psychological consequences such as low self-esteem, social isolation, anxiety, and depression. Among therapeutical approaches, use of Botulinum toxin type A (BoNT-A) on the nonparalyzed side has shown promising results and improvement of quality of life. Nevertheless, the correct technique is paramount, since over-injection of the muscles can result in lack of function, leading to a “paralyzed” appearance, and even worse, functional incompetence, which may cause greater distress to patients. Therefore, the objective of this article is to provide a practical guideline for botulinum toxin use in facial palsy. To this aim, adequate patient assessment, BoNT-A choice, injection plan and dosage, and injection techniques are covered.

## 1. Introduction

Regardless of its etiology, idiopathic (Bell’s Palsy) or secondary facial nerve palsy, due to multiple etiologies such as Ramsey Hunt syndrome, infection, vascular, tumor resection and base of skull injuries, among others, manifests a unilateral peripheral facial nerve palsy which may lead to involuntary static and dynamic alterations of facial expression due to aberrant regeneration of fibers in the neural repair process in as many as 55.5% of patients with longstanding facial weakness [1,2,3,4,5]. The imbalance, static and dynamic, resulting from the muscular paralysis may not only jeopardize simple daily tasks such as articulation, eating and drinking, but is often cosmetically unacceptable to patients due to asymmetry, rendering psychological consequences such as low self-esteem, social isolation, anxiety, and depression [6,7]. The severity of aesthetic changes is related to the absence of movement on the paralyzed side, but also to the response that the mimetic musculature produces after the loss of balance between the paralyzed and moving sides [8,9].

The treatment options for facial nerve palsy usually aim to activate the mimic muscles on the affected side, or improve symmetry on both sides and can range from conservative (pharmacologic and physical therapy) to more invasive approaches (surgical methods, e.g., surgical exploration, decompression or repair depending on the etiology), and the choice depends upon the etiology and the pathogenesis of the condition [1,6]. Among the pharmacologic treatments, botulinum toxin type A (BoNT-A) injection in the nonparalyzed side has been used since 1987 for the treatment of asymmetries caused by facial paralysis and has shown promising results and improvement of quality of life; however, only a small minority of injectors feel able to treat this condition, which requires a complex approach and a detailed knowledge of facial functional anatomy. Botulinum toxin is produced by anaerobic fermentation of the bacterium Clostridium botulinum and consists of the botulinum neurotoxin complexed with a number of neurotoxin-associated proteins, traditionally designated with alphabetical letters from A to H [10,11]. BoNT-A use has expanded for clinical as well as aesthetic indications, and it is now licensed for a broad range of indications, which include, but are not restricted to, temporary improvement of dynamic facial lines, symptomatic relief of blepharospasm, cervical dystonia, various forms of focal spasticity, management of severe hyperhidrosis, sialorrhea and prophylaxis of headaches in adults with chronic migraine [11]. BTX-A blocks acetylcholine release in the nerve endings leading to a reversible muscle paralysis, apparently without long-term damage to the muscle or nerve [11]. Despite the increasing use of botulinum toxin (BoNTA) in treating Facial Palsy, a practical guide for adequate patient assessment, BoNT-A choice, injection plan and injection techniques is lacking. Therefore, the objective of this article is to provide a practical guideline for botulinum toxin use in facial palsy. To this aim we will review patient assessment based on functional anatomy to assist in an individualized approach.

## 2. Facial Palsy

### 2.1. Clinical Manifestation

#### 2.1.1. Flaccid Palsy

Individuals with flaccid paralysis (prosopoplegia) may present absence of wrinkles on the forehead on the paralyzed side and important functional deficits, such as lagophthalmos with potential for corneal ulceration and blindness, as well as oral incompetence, poor articulation, lip and buccal mucosa biting. Furthermore, these individuals develop brow and facial ptosis, severe facial atrophy, an effaced nasolabial fold, and absent animation, affecting mainly the smile mechanism, on the affected side [12,13].

#### 2.1.2. Non-Flaccid Palsy

Non-flaccid facial palsy describes the involuntary, synchronous or overactive movements of facial muscles [14], characterized by the presence of synkinesis and spasms on the paralyzed side. Hyperkinesis is also present on the healthy side [13]. The involuntary static and dynamic changes of facial expression are notably visible during emotional expression, when spontaneous facial movements occur [15].

#### 2.1.3. Synkinesis

Synkinesis is a common and troubling sequelae of facial nerve palsy which can occur in any region of the face [12,16]. Synkinesis refers to synchronous and involuntary movements of certain areas of mimic muscles, which become particularly visible when spontaneous facial movements occur, especially during emotional expressions such as involuntary blinking or smiling [2,15].

Synkinesis physiopathology is very complex and multifactorial and may include aberrant regeneration of facial nerve fibers, ineffective myelination (leading to nerve crosstalk), or a centralized, post injury hyper sensitization of the facial nucleus. Regardless of the proposed mechanism, it may develop during neural repair process three–six months after injury [12,16]. Clinically it can be subtle or completely disfiguring [16,17].

It is traditionally labeled with a composite term, whereby the muscle group of intended movement is followed by the unintended movement muscle group. For instance, “oculo-oral” synkinesis relates to the involuntary movement of the oral commissure secondary to voluntary eye closure.

The oculo-oral (Figure 1) and the oro-ocular remain the most common general forms of synkinesis, with the subject experiencing intense movements of the corner of the mouth with eye closure or spasm of the eye when smiling, respectively [17].Other types of facial synkinesis include chin-oral synkinesis, platysma synkinesis, ocular-chin synkinesis, ocular-nasal synkinesis and chin-ocular synkinesis. The oculo-oral synkinesis may also coexist with other type of synkinesis as a consequence of the involvement of a third muscle with the two initial muscle groups [12,16].

Moreover, simultaneous presence of paretic muscles, synkinetic and hyperkinetic movements can result in abnormal synchronization of facial movements, such as voluntary and reflex activity of muscles that normally do not contract together. The severity of abnormal synchronization correlates with the degree of sequelae of facial palsy [18].

Treatment options for facial synkinesis include musculature retraining with physical therapy, targeted chemo-denervation, and select surgical procedures.

#### 2.1.4. Hyperkinesis

Hyperkinesis consists of static and dynamic asymmetry of the face leading to important functional and aesthetic problems, such as a more pronounced nasolabial fold, deviation of the corner of the mouth laterally up or down and a narrower eye aperture (Figure 2) [2].

This consists of compensatory hyperactivity of the muscles of facial expression on the non-paralyzed side [12,13], against the weak antagonism of the contralateral muscles, leading to the appearance of wrinkles and furrows on the forehead and glabella with an asymmetry of brow positioning, deviation of the nose and the mouth to the hyperactive side, and hyperactivity of the depressor labii inferiors and of upper lip-elevator muscles [4,13].

#### 2.1.5. Spasms

Spasms can be observed in the post paralytic facial syndrome as a consequence of a significant percentage of axons’ Wallerian degeneration. This facial motor dysfunction may be spontaneous or secondary to involuntary contraction of facial muscles on the paralyzed side [13,19]. It can be triggered by automatic or emotional facial movements resulting in a very conformable mass of contractions, sometimes leading to emotional distress and depression. Clinically it is more frequently characterized by rapid and rhythmic contractions that may resemble a “tick” but can also manifest as a sustained muscle contraction. The spasm reported by facial palsy patients may mimic those observed in essential hemifacial spasm; however, they are clinically and neuro-physiologically distinct. Usually, the spastic muscle is associated with synkinesis [13,19].

## 3. Patient Evaluation

A comprehensive evaluation of the history of facial palsy is critical, with particular attention to the time-course of the disorder, as well as any contraindications to BoNT-A injection. It is controversial whether BoNT-A within 6 months after the onset of facial paralysis should be avoided, due to a possible risk of worsening of the synkinesis [12,17]. However, a study published by Kim showed a potential benefit in the treatment of acute facial palsy (from one to six months after initial onset) using BoNT-A injection, improving symmetrical function by decreasing contralateral hyperkinesis [20]. Moreover, the patient should be questioned about his main concerns and which areas of the face and neck are the most troubling, since these areas should be the first to be addressed.

Furthermore, a thorough understanding of the facial musculature and neural anatomy, its role in facial expression, as well as its function, is paramount to determine the required injection areas, definition and distribution of injection points [12]. The facial nerve is responsible not only for stimulating the mimetic muscles, creating a balance between the synergic and antagonist forces, but also for maintaining the muscular tonus in the relaxed state and for voluntary and involuntary muscle contraction [9]. In facial paralysis the imbalance of sectoral forces creates facial deviation that can be observed at rest and in the dynamic state [21].

Facial muscles or muscles of facial expression are usually thin, delicate and concealed by fat. During facial muscles contraction, displacement of the overlying skin, fat pads, and even other facial muscles may occur, and repetitive movements may lead to wrinkles, furrows, edges and bulges. The combination of these changes is called facial expression [22]. Dynamic wrinkles appear as lines in the skin overlying the contracted muscle, always perpendicularly to the muscle fibers, enabling inference of which muscle is involved in every wrinkle. For instance, the procerus pulls the skin of the glabellar region down and creates a horizontal line across the bridge of the nose, perpendicularly to its fiber direction [22]. Summary of the facial expression muscles’ action and criteria for treatment can be seen in Table 1.

Muscles should be carefully assessed at rest and at maximum contraction, while observing the patient speaking and performing specific facial movements. Standardized photography and video recording improve assessment. The nine standard views include: face at rest, brow elevation, complete eye closure, nose wrinkling, grin, pucker, pout, whistle and lower lip depression (Figure 3) [12,17]. It is also crucial during the evaluation to identify the “triggers” for eliciting synkinesis. The most common are a forceful eye closure or a puckering movement of the lips. An individualized injection plan should subsequently be developed, with definition of injection points, as well as dose per point to be injected, having in mind that the main objective is to balance the facial expression without freezing, avoiding overtreatment [12].

Extra care should be taken while treating patients with severe chronic facial palsy presenting multiple synkinesis, to whom BoNT-A injection could cause further weakness of paretic muscles instead of reduction of synkinetic movements [16].

### Scales for Facial Palsy Grading

To determine the severity of facial paralysis, a facial nerve grading system is often used within clinical practice. Physicians should select a preferential system and document consistently. The House-Brackmann facial nerve grading system (HB) and the Sunnybrook facial grading system (SB) are the most frequently used. The HB grades facial function between I and VI, with grade I being normal facial function and grade VI being total paralysis [23]. The SB system is based on the evaluation of symmetry at rest, symmetry of voluntary movement, assessing the degree of voluntary excursion of facial muscles, and synkinesis, rating the degree of involuntary muscle contraction associated with each facial expression [24]. This scale indicates the level of facial dysfunction grading from 0 (complete palsy) to 100 (complete recovery), and allows a separate analysis of the synkinesis, grading the severity using a score from 0 to 15.

A validated Synkinesis Assessment Questionnaire (SAQ) is also used in routine patient care as a clinical tool and for making treatment decisions. The nine questions are answered on a scale from 1 to 5, where 1 corresponds to seldom, or not at all, and 5 to all the time or severely [25].

## 4. Botulinum Toxin in Facial Palsy Approach

Among the conservative approaches for facial palsy, botulinum toxin injection with or without neuromuscular therapy has shown promising results as a non-operative method in restoring normal facial features [17]. The use of botulinum toxin A injection has more recently gained acceptance as the appropriate management for patients affected with this injury, since the practicability of injection and flexibility of injection points allow an individualized and customized treatment. Moreover, improvement is observed within a few days, and the effect may be long lasting. The efficacy of BoNTA in reducing facial asymmetry in the short-term and the reduced rate of late complications including contractures and synkinesis has been demonstrated in recent clinical trials [26].

### 4.1. Technique

In order to create more symmetry at rest and with animation, BoNT-A is injected in targeted muscles of the unaffected side to reduce hyperkinesis, resulting in a significant aesthetic improvement of the face. For the treatment of synkinesis, BoNT-A injection into a specific muscle can reduce or eliminate the involuntary muscle action that is aberrantly triggered [12]. Nevertheless, synkinetic movements involving the Depressor Anguli Oris (DAO) and Zygomatic muscles may not present an adequate response to BoNT-A injection, with partial efficacy and small duration of effect, probably due to the huge strength of DAO [13].

### 4.2. Injection Technique

The injection technique needs also be taken into account in order to reduce adverse events and optimize efficacy. The precise injection area, as well as depth and angle of the needle while injecting, are paramount, since muscular-skin units may differ in terms of thickness. For instance, even though the injection points for blocking the action of the *corrugator supercilii* and the *depressor supercilii* origin are the same, the depth of injection differs, needing to be close to the bone and subdermal, respectively [14]. Moreover, the direction of the needle while injecting is also important to prevent side effects, including diplopia, ptosis and paralysis of unwanted muscles. Further details, such as origin and insertion of the facial expression muscles, which is generally from the skull into the superficial fascia, dermis or even other muscles, may influence the depth of injection, this being more profound at the origin of the muscle and more superficial at the muscle insertion [22]. Detailed information on injection technique (e.g., depth of the needle tip, direction of needle) can be observed in Table 2.

### 4.3. Different BoNTA Formulations

Understanding the diffusion properties of the chosen BoNTA is crucial, since different botulinum toxins may have distinct characteristics. Currently, several botulinum toxin type A formulations are approved for aesthetic and therapeutic use, the most widely used of which include Onabotulinum-toxin A (ONA; Botox^®^/Vistabel^®^, Allergan Inc., Irvine, CA, USA), Abobotulinum-toxin A (ABO; Dysport^®^/Azzalure^®^, Ipsen, Paris, France), and Incobotulinum-toxin A (INCO; Xeomin^®^/Bocouture^®^, Merz Pharmaceuticals GmbH, Frankfurt am Main, Germany). Though all these formulations have the same mode of action and each contain the core 150 kDa neurotoxin, INCO remains the only formulation which is purified to contain just the required therapeutic component and is free from unnecessary bacterial proteins [27].

Among the favorable pharmacological properties that have made BoNTs unique drugs is the limited diffusion when locally injected [28]. Several factors influence diffusion, including preparation characteristics, volume, dosing, injection technique, and muscles injected [18,29,30]. ONA and INCO have comparable efficacies with a 1:1 conversion ratio and have demonstrated therapeutic equivalence in different indications including cervical dystonia, blepharospasm and aesthetic. An ONA to ABO conversion ratio ≤ 1:3 should be considered the most appropriate [11]. Recently a study demonstrated that there is no difference in light chain activity between ONA, INCO and ABO; however, at the FDA-approved doses/conversion rate, a greater amount of neurotoxin is injected with ABO than with other BoNT-A products [31]. When using a conversion ratio of 2.5:1 (ABO:INCO/ONA, respectively), the maximal area of anhidrosis of ABO seem to be greater than that of INCO and ONA [27,32]. Mice injected with INCO showed a volume-dependent reduction in wheel running, with larger volumes inducing more intense paresis [30]. Moreover, in the treatment of idiopathic detrusor hyperactivity, the injected volume was a key factor in the spread and action of BoNT/A in the bladder, with a wider diffusion of the toxin when higher volumes were injected [33]. This evidence indicate that the diffusion is dependent on injected volume and the total amount of neurotoxin, taking into account the size of the muscle/area to be injected [28]. Understanding the action halo of different BoNT-A preparations is of crucial clinical relevance, since precise localization of the clinical effect is essential to avoid adverse events such as the paralyzing effect of a closely positioned adjacent muscle of facial expression. A larger action halo may lead to unwanted outcomes, such as ptosis, heavy brow, or a frozen face in terms of facial aesthetics [32]. A systematic review and meta-analysis of randomized clinical trials did not find reports of serious side effects due to BoNT treatment and indicated that the overall rate of adverse events, from mild to moderate, was 25% in BoNT-A-treated subjects, compared with 15% in the placebo group [34]. Adverse events were observed in 9.8% of the patients with facial nerve palsy after neurosurgery treated with incobotulinum-toxin A injection [26], whilst Do Nascimento Remiglio et al. observed a higher incidence of excessive facial weakness in the group treated with abobotulinum-toxin A group when compared with the onabotulinum-toxin A group (93.3% vs. 64%, respectively) [14].

### 4.4. Dosage

Doses and points of injection may vary individually according to the severity of synkinesis or hyperkinesis, as well as the muscle involved [15]. It is recommended to inject lower-than-normal amounts at the initial treatment session, with a two-week follow-up for possible additional injections, since this allows a better understanding of the patient’s BoNT-A requirements. Furthermore, BoNT-A products show a direct, proportional relationship between the halo action size and the number of units injected [35].

After several sessions, a customized dose and points of injection scheme can be developed [12]. The ideal injection framework, as well as mean doses needed to restore facial symmetry for each muscle group remain empirical. Dose ranges suggested for the injection for each muscle of facial expression in Table 2 were based on previous review studies of BoNT-A use in facial palsy [12,13,14,15,17,26] and BoNT-A in aesthetic medicine [36,37,38].

### 4.5. Points of Injection

Points of injection usually performed for the treatment of hyperkinesis and synkinesis, and dosage of BoNT-A per injection point are detailed in Figure 4 and Table 2.

#### 4.5.1. Upper Face

While treating the upper face it is important to maintain a natural appearance and a balance between the elevator and depressor muscle groups: the *frontalis* is the only muscle that elevates the eyebrow, whilst *corrugator supercilli*, *procerus*, *depressor supercilli* and the *orbicularis oculi* are considered depressor muscles. In order to balance this compensatory hyperactivity on the non-paralyzed side, high doses should be injected into the frontalis muscle and the brow depressors (Figure 2) [13]. While defining the distribution of injection points and BoNT-A dose, palpation of the forehead while the patient contracts the frontalis allows a better assessment of the presence or absence of muscle contraction, as well as of muscle strength [38].

#### 4.5.2. Mid-Lower Face

The mid and lower face are the most challenging areas to be treated. Over-injection of the muscles can result in lack of function of the muscles related to the smile, leading to a “paralyzed” appearance, and even worse, oral incompetence, which may cause a greater distress to patients [17].

Assessment of the muscles that control the movements of the lips and cheeks is the most complex. The muscles that raise the lips comprise the *elevator labii superioris*, the *elevator anguli oris*, the major and minor zygomatic muscles, and the *elevator labii superiors alaeque nasi*. On the contrary, the muscles that act upon the lower lip include the *depressor labii inferiors*, the *depressor anguli oris*, the *mentalis* and to a lesser extent the platysma. With the exception of the *mentalis*, they all interdigitate and blend closely with the *orbicularis oris* (Figure 1 and Figure 5*)* [9]. Hyperactivity of the zygomatic major and minor muscles elevate the oral commissure, whilst the *depressor anguli oris* and the *depressor labii inferioris* depress the lip, leading to an exaggerated asymmetry during smiling or laughing. It is important to differentiate the action of the *depressor labii inferioris* and the *depressor anguli oris* in the downward movement of the inferior lip, in order to define whether both muscles must be treated or only one of them. The same rationale should be used while assessing the upward movement of the superior lip. A detailed description of muscles’ action can be seen in Table 1.

It is recommended to inject lower doses for the treatment of the Mid-Lower face at the first treatment session, targeting muscles with predominant contraction, with a revising session after two weeks for possible additional injections sites and/or increase of BoNT-A dose. After some sessions have taken place, a customized plan for injection sites and doses can be defined.

Tense spasms of the platysma muscle on the affected side are a frequent complaint among synkinesis patients. The medial fibers of the platysma contribute to the depressor function of the lateral oral commissure, since it pulls the corner of the mouth in a downward direction, against the zygomatic muscles, which on the other hand provide an upward and lateral excursion of the oral commissure. As a result, after treating the platysma muscle with BoNT-A patients usually report improvement in their ability to smile (Figure 5D–F). Therefore, the platysma and *depressor anguli oris,* regions typically affected, need to be aggressively injected, to reduce banding and tightness in the neck and improve the smile mechanism [12,17].

#### 4.5.3. Adverse Events

Adverse events occur more frequently related to the diffusion of the neurotoxin beyond the target muscle, and may be due to higher BoNT-A doses, inadequate conversion ratio, large volume injected or technical issues [32]. In order to minimize adverse events related to dose, de Maio et al. suggested that a lower dose treatment, with a second treatment after two weeks, may minimize the risk [21]. The precise point and depth of injection and direction of the needle are also important. Another potential issue is BoNT-A over treatment causing cosmetic, expressional and functional deficiencies [12]. Clinical consequences will vary according to the injection site and comprise oral incompetence, speech abnormalities, diplopia, ptosis, lagopthalmos, worsening of cosmetics and dysphonia [12,13]. Adverse events around the mouth tend to be worse at day 14 and decrease with time as the effect of the toxin reduces and the patient adapts to the new dynamics [9,14].

## 5. Conclusions

Botulinum toxin injection is effective in restoring facial symmetry, reducing hyperkinesis, synkinesis and facial imbalance due to facial palsy. A development of an individualized injection plan after a thorough patient evaluation based on functional anatomy, as well as adequate choice of injection technique, leads to reduction of adverse events, optimizing efficacy and patient satisfaction.

## Figures and Tables

**Figure 1 toxins-13-00159-f001:**
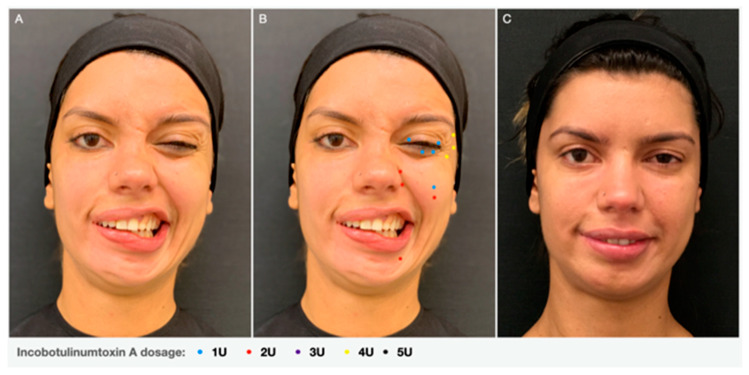
Oculo-oral synkinesis: (**A**) voluntary eye closure on the left and an involuntary movement of the oral commissure on the non-paralyzed side due to aberrant activity of the buccinator muscle. (**B**) Injection points: Lateral portion and pre-tarsal region of Orbicularis oculi; Depressor labii inferioris and lip elevators—zygomatic major and minor, levator labii superioris and levator labii superioris alaeque nasi. (**C**) Clinical result 15 days after Botulinim toxin type A (BoNT-A) injection.

**Figure 2 toxins-13-00159-f002:**
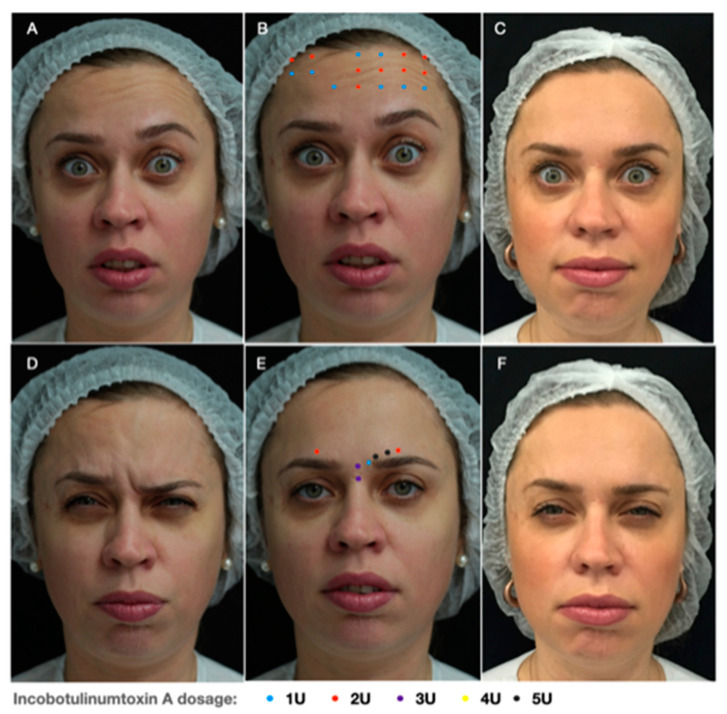
Hyperkinesis: A 36-year-old-patient presenting (**A**) hyperkinesis of the frontalis muscle and (**D**) brow depression on the left, showing a balance of the eyebrow positioning and elimination of wrinkles while contracting. Injection points scheme and related doses of Incobotulinum-toxin A in the (**B**) forehead and (**E**) glabella. Clinical result 15 days after BoNT-A injection (**C**,**F**).

**Figure 3 toxins-13-00159-f003:**
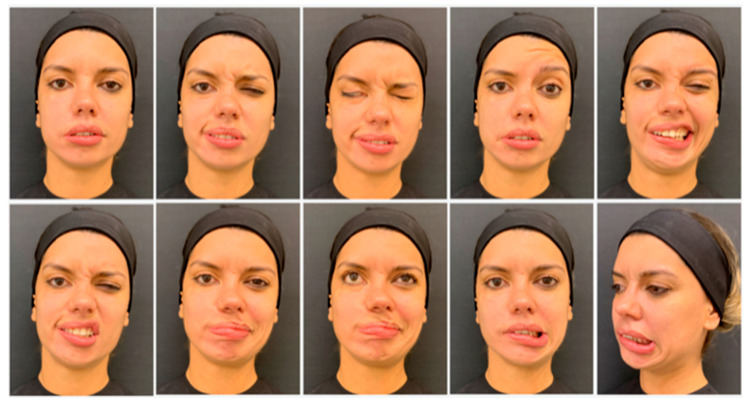
The nine standard frontal views (beginning at top row, on the far left, respectively): face at rest, pucker, complete eye closure, brow elevation, grin, nose wrinkling, whistling, pouting and lower lip depression, and a profile view for the assessment of the platysma action.

**Figure 4 toxins-13-00159-f004:**
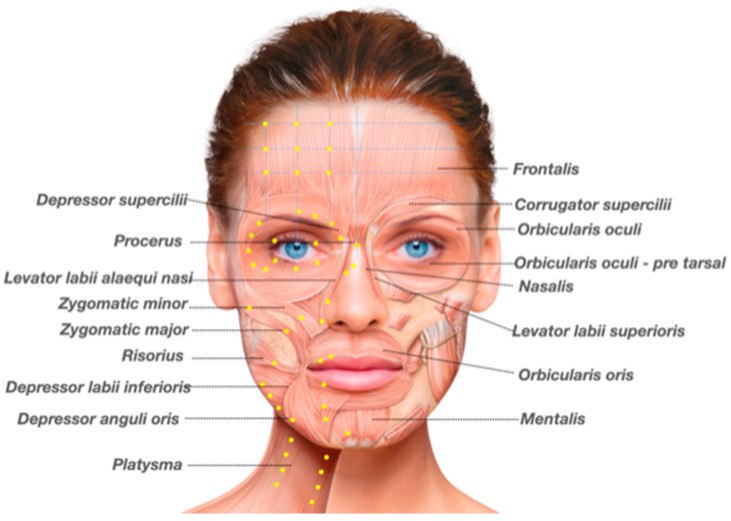
Possible injection sites for botulinum toxin in the nonparalyzed side, according to the facial expression muscles to be assessed: frontalis, corrugator supercilii, procerus, depressor supercilii, orbicularis oculi pars orbicularis, orbicularis oculi pars tarsalis, nasalis, zygomatic major, zygomatic minos, levator labii superioris, levator labii superioris alaeque nasi, risorius, depressor anguli oris, depressor labii inferioris, mentalis, and platysma. (Image reproduced from Merz Institute Advanced Aesthetics platform—www.merz-institute.com).

**Figure 5 toxins-13-00159-f005:**
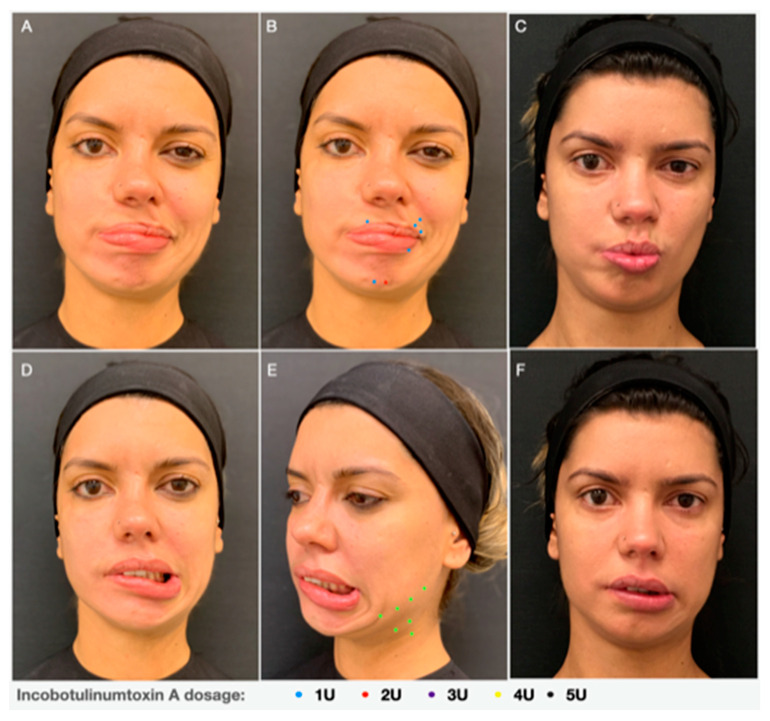
A 27-year-old female with facial palsy since seven years old, treated with Incobotulinum-toxin A injections on the non-paralyzed side (**A**,**D**). Points distribution scheme of INCO injection and doses in the (**B**) *orbicularis oris* and *mentalis* and (**E**) platysma. Clinical result 15 days after INCO injection (**C**,**F**).

**Table 1 toxins-13-00159-t001:** Facial Muscles’ actions and criteria for assessment.

Muscle Group	Action	Criteria for Treatment
Occipito-Frontalis	Moves de scalp back, raises the eyebrow and wrinkles the forehead	Asymmetry in eyebrow position. Frontal wrinkles (horizontal lines in the forehead) at rest on the nonparalyzed side
Corrugator supercilii	Pulls the eyebrows downward and toward the midline of the nose	Vertical medial lines in the glabellar region on the nonparalyzed side. Asymmetry in eyebrow positioning
Procerus	Draws down the skin between the eyebrows and assists in flaring the nostrils	Horizontal lines located at the root of the nose
Depressor supercilii	Draws down the eyebrows	Diagonal lines medially to the medial end of the eyebrow
Orbicularis oculi	Responsible for closing the eyelids, blinking and development of crow’s feet	Periorbital wrinkles in contraction and at rest, and pre-tarsal hypertrophy in the nonparalyzed side. Asymmetry in the eye opening.
Nasalis	Compresses the nostrils and may completely close them	Asymmetry of nasal wrinkles when sniffing
Levator labii superioris alaeque nasi	Dilates the nostril and elevates the lateral upper lip and wing of the nose	Asymmetry of the medial third of the nasolabial fold when sniffing. Center of the upper lip pulled up. Deepening of the nasolabial fold
Levator labii superioris	Elevates and everts the upper lip	Asymmetry of the medial third of the nasolabial fold. Exposure of the superior dental arch while smiling
Levator anguli oris	Pulls the angle of the mouth straight up and curves the mouth line upward at its ends	Asymmetry in the positioning of the lip corner
Zygomaticus Major	Pulls the angle of the mouth up out and backwards when smiling	Smile asymmetry in the upper lateral direction of the non-paralyzed side
Zygomaticus Minor	Pulls the middle section of the nasolabial furrow and middle portion of the side of the upper lip outward and slightly upward	Lip pulled upward and laterally, deepening of the upper middle portion of the nasolabial furrow
Orbicularis oris	Kissing, puckering and pressing lips against the teeth, closing the mouth	Philtrum deviation to the nonparalyzed sideAsymmetry of perioral wrinkles
Risorius	Pulls the modulus and therefore the angle of the mouth out and backwards	Lateral asymmetry at retraction of the angle of the mouth (sardonic expression)
Mentalis	Raises and wrinkles the skin of the chin, thus elevating the lower lip	“orange peel” texture and deep transverse mental crease on the nonparalyzed side
Depressor labii inferioris	Pulls the middle third of the lower lip straight downward	Smile asymmetries. Exposure of lower teeth arch on the non-paralyzed side when smiling. Lip asymmetry when ‘‘showing lower teeth’’
Depressor anguli oris	Depresses and lateral displace the angle of the mouth	Corner of the lips pulled down, Deepening of the nasolabial furrow
Platysma	Pulls the lower lip and corner of the mouth sideways and downwards	Excessive movement of the nonparalyzed side platysma when ‘‘showing lower teeth’’

**Table 2 toxins-13-00159-t002:** Injection points, doses and suggested technique.

Muscle Group	Points of Injection	Dosage	Injection Technique *
Occipito-Frontalis	3 to 9	0.5 to 2 U/point	Insert 30% of the needle angled upward. Extend the injection sites laterally along the lateral canthus
Corrugator supercilii	2 (origin and tail) * Third point (lateral, blending with orbicularIs oculi fibers)	4 to 6 U/point * 3rd point: 0.5 to 2 U	Origin: introduce 100% of the needle Insertion: introduce 30% of the needle * Third point: introduce the needle bevel only
Procerus	1 to 2	4 to 8 U	Introduce 50% of the needle directing upwards
Depressor supercilii	1	1 to 2 U	Subdermal injection, directing the needle medially and upwards
Orbicularis oculi	3 to 6	4 U/point (lateral radial lines) 2 U/point (inferior line and lateral end of the brow)	Insert 30% of the needle superficially, directing the needle away from the eye
Pre-tarsal	1 to 5 points	0.5 to 1 U/point	Intradermal injection 3 mm below the ciliary margin
Nasalis	1 to 2	2 to 4 U/point	Intramuscular injection with needle at 45 ° to the skin, on the superior third of the lateral nasal wall, inferior to the angular vein
Levator labii superioris alaeque nasi	1	2 U	Naso-facial groove at the superior point of the nasal ala. Insert 50% of the needle pointed upward
Levator labii superioris	1	1–2 U	Injection with needle perpendicular to the skin, lateral to the bony nasal prominence. Insert 50% of the needle pointed upward
Zygomaticus Major	1 to 2	1 to 2 U/point	Superficial injection (50% of the needle, perpendicular to the skin): first point 2 cm from the oral commissure on a diagonal line to the malar prominence. second point on the malar prominence
Zygomaticus Minor	1	2 U	Superficial injection (50% of the needle, perpendicular to the skin) 1 cm above the zygomaticus major
Orbicularis oris	2 to 6	1 U/point	Superficial injection at or 2 mm above the vermilion border. Insert only the needle bevel pointed upward
Risorius	1	1 to 2 U/point	Deep injection (100% of the needle, perpendicular to the skin) 2 cm medially to the anterior border of the masseter
Mentalis	1 to 2	2 to 4 U/point	Deep injection (100% of the needle, perpendicular to the skin) on the chin prominence, close to the midline, at a distance >2 cm from the lower lip
Depressor labii inferioris	1 to 2	1 to 3 U/point	Superficial injection (30%–50% of the needle) 1 to 1.5 cm from lower lip border, 1 cm from the midline
Depressor anguli oris	1	2 to 4 U	Superficial injection (50% of the needle, perpendicular to the skin) immediately above the angle of the mandible and 1 cm lateral to the oral commissure
Platysma (bands and jawline)	6 to 10	2 U/point	Superficial intramuscular injections, 1.5 to 2 cm apart along the mandible margin and platysma bands.

* Injection technique has been taken into account using a 6 mm needle.

## Data Availability

The data presented in this study are available on request from the corresponding author. The data are not publicly available due to privacy.

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
