# Peer review of "Botulinum Toxin Type A to Improve Facial Symmetry in Facial Palsy: A Practical Guideline and Clinical Experience"

_toxins, 2021, doi:10.3390/toxins13020159_

Round 1
Reviewer 1 Report
Dear Authors,
Despite potentially interesting, your paper content is not good enough to be published.
The aim of the study is unclear in the text, despite clearly defined in the title.
There are too much info for a scientific article, and it seems me more a book chapter than a scientific paper.
You didn't report the method with which you selected the articles included in your review, which in my opinion are too few. In fact, you completely omitted a series of studies which analyzed the use of BTX.
You didn't mention any type of methods used to evaluate the facial asymmetry, instead you described a clinical evaluation without any scientific criteria.
Your conclusions are not supported by your study, that it seems more a dissertation rather than an article.
A review have to be focused on clarify an aspect, or to reinforce a concept that is doubt, you didn't any of both.
I added some comments in the PDF.
My suggestion is to re-write the article following PRISMA guidelines for review, specify a clear aim and remain focused on it without adding to much info, which are deleterious for readers' understanding.

Author Response
Dear Reviewer,
Thank you very much for your valuable comments and inputs, which we´ve taken into very careful consideration.
Even though BONT-A is widely used for the treatment of facial nerve palsy, the dosage, evaluation and technique are far from a consensus. Thus, the aim of the article would be to provide a more practical approach of facial palsy for the attending physician, rather than a review of the literature regarding this theme.
As such, we agree that the title could be misleading and besides having improved the title and aim of the article, we've removed the excess of information and reorganised the topics so that to improve readability.
Best regards

Reviewer 2 Report
Page 4 Line 106-108 It should be mentioned that there are serotypes A to G of botulinum toxin. Botulinum toxin type A (BTX-A) is licensed for the indications listed below in the text
110 Indications: Sialorrhea should be added
113 ... blocks acetylcholine release from nerve endings
115 Reference [4] is not approriate please replace e.g. current review about action mechanism and pharmacology of BoNT
127 practallity better: practicabillity
129 improvement is not observed promptly, takes a couple of days
Page 10 207 It should be mentioned that the greater halo of ABO might be the result of the conversion ratio and not a different characteristic of ABO (because the neurotoxin is the same in all products)
109/209 Adverse events are only briefly mentioned in the text. Some short information should be amemded related to possible events like eylid or eybrow ptosis .... Adverse events are determined by the dose and injection volume (maybe added in line 218)
Author Response
Dear Reviewer,
Thank you very much for your most valuable comments!
I´ve taken the liberty to copy the comments below, in order to make sure each one is addressed properly:
Comment 1: Page 4 Line 106-108 It should be mentioned that there are serotypes A to G of botulinum toxin.
Response: Thank you for the comment. Agreed and added the information (Line 50)
Comment 2: Botulinum toxin type A (BTX-A) is licensed for the indications listed below in the text
110 Indications: Sialorrhea should be added
Response: Thank you for the comment. Sialorrhea was added as an indication (Line52-54)
Comment 3: 113 ... blocks acetylcholine release from nerve endings Reference [4] is not approriate please replace e.g. current review about action mechanism and pharmacology of BoNT
Response: Thank you for the comment. Reference was replaced accordingly( line 50).
Comment 5: 127 practallity better: practicabillity
Response: Thank you for the comment. Agreed and corrected (Line 198)
Comment 6: 129 improvement is not observed promptly, takes a couple of days
Response: Thank you for the comment. Replaced promptly by " within a few days" ir order to avoid misunderstanding (Line 200).
Comment 7: 207 It should be mentioned that the greater halo of ABO might be the result of the conversion ratio and not a different characteristic of ABO (because the neurotoxin is the same in all products)
Response: Thank you for your comment. We’ve mentioned in the text the factors influencing action halo (lines 246-262)
Comment 8:109/209 Adverse events are only briefly mentioned in the text. Some short information should be amemded related to possible events like eylid or eybrow ptosis .... Adverse events are determined by the dose and injection volume (maybe added in line 218)
Response: Thank you. We’ve inserted a more comprehensive text regarding Adverse Events (lines 334-344 )
Reviewer 3 Report
It is better to give the rewiew of the literature in the beginning.
The terms "flaccid palsy" and "non-flaccid palsy" are not correct to describe the clinical variant of facial palsy. Terms Prosopoplegia and Prosopoparesis are more correct.
Evaluation of patients is better for understanding with using of standart scales (House-Brackmann, scales of synkinesis and QOL).
I cannot agree with the recommednation to avoid BTX-injection within 6 months after the onset of facial palsy. In contrast, in cases of acute prosopoplegia injection in mimic muscles of the "healthy side" leads to better regress of paresis and do not lead to a worsening of the synkinesis
Author Response
Dear Reviewer,
Thank you so much for your most valuable comments!
I´ve taken the liberty to copy your comments below, so that I can properly address each:
Comment 1:It is better to give the rewiew of the literature in the beginning.
Response: Thank you for the suggestion. We’ve adapted the text bringind the review section of the paper to the beginning and reordered the sections accordingly.
Comment 2: The terms "flaccid palsy" and "non-flaccid palsy" are not correct to describe the clinical variant of facial palsy. Terms Prosopoplegia and Prosopoparesis are more correct.
Response: Thank you for your comment. The correct terms were added accordingly (Line 63).
Comment 3: Evaluation of patients is better for understanding with using of standart scales (House-Brackmann, scales of synkinesis and QOL).
Response:Thank you for you comment. We've added the scales to the text (Lines 179-193)
Comment 4: I cannot agree with the recommednation to avoid BTX-injection within 6 months after the onset of facial palsy. In contrast, in cases of acute prosopoplegia injection in mimic muscles of the "healthy side" leads to better regress of paresis and do not lead to a worsening of the synkinesis
Response:
Reviewer 4 Report
The manuscript is well written and worthy for publication in the present status.
I think that the published paper will help physicians in the treatment of facial palsy.
Authors should check for the titles of the 2.1 and 2.2 paragraphs as well as for the figure 3 caption.
Author Response
Dear Reviewer,
Thank you very much for your review and comments.
Titles of 2.1 and 2.2 paragraphs were corrected accordingly ( lines ) and caption for figure 3 was clarified.
Reviewer 5 Report
The manuscript deals with the use of botulinum toxin to improve facial symmetry in facial palsy. Although it is well-structured and logically sound, it is difficult to find differences from previous studies. Therefore, it is necessary to describe what makes this manuscript different from other studies.Author Response
Dear reviewer,
Thank you very much for your comments. The main idea of this article would be to provide a practical guide for Facial Palsy treatment, focusing on patient assessment based on functional anatomy, considerations on BoNT-A choice and dose, as well as injection plan and injection techniques in order to develop a customized treatment approach and reduce injection risks.
Best regards
Round 2
Reviewer 1 Report
You did not address my comments. I cannot accept your manuscript
Reviewer 5 Report
Dear Author,
Thank you for your quick response to comments on the thesis and revising them. After reviewing your manuscript again, I confirm that you have followed the advice given and that the manuscript is well-organized and worthy of publishing.
I am pleased to be offered the opportunity to review this paper.
Thank you.